# YcgC represents a new protein deacetylase family in prokaryotes

Shun Tu[1,2†], Shu-Juan Guo[1,2†], Chien-Sheng Chen[3†], Cheng-Xi Liu[1,2], He-Wei Jiang[1,2], Feng Ge[4], Jiao-Yu Deng[5], Yi-Ming Zhou[6], Daniel M Czajkowsky[7], Yang Li[1,2], Bang-Ruo Qi[1,2], Young-Hoon Ahn[8], Philip A Cole[8*], Heng Zhu[8,9*], Sheng-Ce Tao[1,2,7*]

[1]Shanghai Center for Systems Biomedicine, Key Laboratory of Systems Biomedicine, Shanghai Jiao Tong University, Shanghai, China; [2]State Key Laboratory of Oncogenes and Related Genes, Shanghai, China; [3]Graduate Institute of Systems Biology and Bioinformatics, National Central University, Jhongli, Taiwan; [4]Key Laboratory of Algal Biology, Institute of Hydrobiology, Chinese Academy of Sciences, Wuhan, Hubei, China; [5]State Key Laboratory of Virology, Wuhan Institute of Virology, Chinese Academy of Sciences, Wuhan, China; [6]National Engineering Research Center for Beijing Biochip Technology, Beijing, China; [7]Bio-ID Center, School of Biomedical Engineering, Shanghai Jiao Tong University, Shanghai, China; [8]Department of Pharmacology and Molecular Sciences, Johns Hopkins University School of Medicine, Baltimore, United States; [9]The High-Throughput Biology Center, Johns Hopkins University School of Medicine, Baltimore, United States

**\*For correspondence:** pcole@jhmi.edu (PAC); hzhu4@jhmi.edu (HZ); taosc@sjtu.edu.cn (S-CT)

[†]These authors contributed equally to this work

**Abstract** Reversible lysine acetylation is one of the most important protein posttranslational modifications that plays essential roles in both prokaryotes and eukaryotes. However, only a few lysine deacetylases (KDACs) have been identified in prokaryotes, perhaps in part due to their limited sequence homology. Herein, we developed a 'clip-chip' strategy to enable unbiased, activity-based discovery of novel KDACs in the *Escherichia coli* proteome. In-depth biochemical characterization confirmed that YcgC is a serine hydrolase involving Ser200 as the catalytic nucleophile for lysine deacetylation and does not use $NAD^+$ or $Zn^{2+}$ like other established KDACs. Further, in vivo characterization demonstrated that YcgC regulates transcription by catalyzing deacetylation of Lys52 and Lys62 of a transcriptional repressor RutR. Importantly, YcgC targets a distinct set of substrates from the only known *E. coli* KDAC CobB. Analysis of YcgC's bacterial homologs confirmed that they also exhibit KDAC activity. YcgC thus represents a novel family of prokaryotic KDACs.

## Introduction

Protein (de)acetylation plays critical roles in many key biological processes, for example, transcriptional regulation, aging, and metabolism (*Cohen et al., 2004*; *Grunstein, 1997*; *Lin et al., 2009*; *Lu et al., 2011*). Recent mass spectrometry (MS) efforts have revealed that many proteins are acetylated in *Escherichia coli*, although only a single *E. coli* lysine deacetylase (KDAC), CobB, has been identified so far (*Choudhary et al., 2009*; *Henriksen et al., 2012*; *Zhang et al., 2013a*). The fact that induction of CobB only had a limited impact on reducing the global protein acetylation level suggests that additional KDACs may exist. However, homolog searching has failed to reveal any additional KDACs in *E. coli*, presumably because these enzymes emerged via convergent evolution. In contrast to bioinformatics methods, biochemical approaches have proven effective for identifying

**eLife digest** After proteins have been made, they can be modified in several ways. For example, chemical tags called acetyl groups may be added to (and later removed from) the protein to regulate cell activities such as aging and metabolism. Enzymes are proteins that help catalyze the reactions that add or remove the acetyl tags on certain "substrate" proteins. In the bacteria species *Escherichia coli*, many enzymes that help to add acetyl groups to proteins have been discovered. However, only a single *E. coli* "deacetylase" enzyme that removes the acetyl group has been identified.

Now, Tu, Guo, Chen et al. have devised a technique to identify new deacetylases, called the "clip-chip" approach. In this method, thousands of proteins that are potential deacetylases are arrayed on a glass slide, and substrate proteins are immobilized on another slide. The two slides are then clipped together face-to-face, allowing the potential enzymes to transfer to the substrate slide and interact with the substrates.

Using this approach, Tu, Guo, Chen et al. identified a protein called YcgC as a deacetylase in bacteria. Further characterization experiments revealed that YcgC works in a different way to other known deacetylases, and that it targets different substrates to the previously known *E. coli* deacetylase.

Tu, Guo, Chen et al. found that the equivalents of YcgC in other bacteria species are also deacetylases; these enzymes therefore represent a new deacetylase family. In the future, the clip-chip approach could be used to discover new members of other enzyme families.

new enzymes resulting from convergent evolution (*Tsukada et al., 2006*; *Yamane et al., 2006*), though their laborious, time-consuming nature has limited their applications to high-throughput, proteome-wide screens. Herein, we established a 'clip-chip' approach to enable a proteome-wide, activity-based search for novel KDACs in *E. coli*.

## Results

### The clip-chip strategy

The principle behind the clip-chip approach is the delivery of thousands of purified proteins spotted on a glass slide (e.g., a proteome microarray) to a substrate of interest immobilized on another slide (i.e., the substrate slide) such that thousands of desired biochemical reactions can be carried out in parallel, in order to identify new enzymes of interest (*Figure 1a* and *Figure 1—figure supplement 1*). The substrate slide is created by immobilizing a substrate of interest onto a nitrocellulose-coated slide. After thousands of purified proteins are spotted on a plain glass slide, it is then 'clipped' onto the substrate slide in a face-to-face manner, resulting in the delivery of the proteins onto the substrate slide. Owing to the highly porous nature of nitrocellulose and the tiny volume of the protein droplets (0.3–0.5 nL), the delivered protein droplets are immediately absorbed and kept locally in the nitrocellulose, preventing cross-contamination. To determine which transferred proteins possess the enzymatic activity in question, the 'clipped' substrate slide is then incubated with an appropriate reaction buffer, followed by signal detection.

### Screen new KDAC using the *E. coli* proteome microarray

To screen for new KDAC candidates in the *E. coli* proteome, we prepared separate substrate slides for three *E. coli* proteins, namely NhoA, RutR, and YceC, which were chosen because they have a rather high endogenous acetylation level and because CobB exhibits only modest ability to deacetylate them (*Zhang et al., 2013b*). After 4256 individually purified *E. coli* proteins (*Chen et al., 2008*) were spotted on plain glass slides, they were clipped separately onto the three substrate slides, followed by incubation with a standard deacetylase reaction buffer containing NAD⁺. The reactions were terminated by adding wash buffers, followed by a signal detection step with a pan α-acetyllysine (α-AcK) antibody coupled with Cy3-labeled secondary antibodies as detection reagents. Proteins that efficiently deacetylated the substrates could be readily identified as they left behind pairs

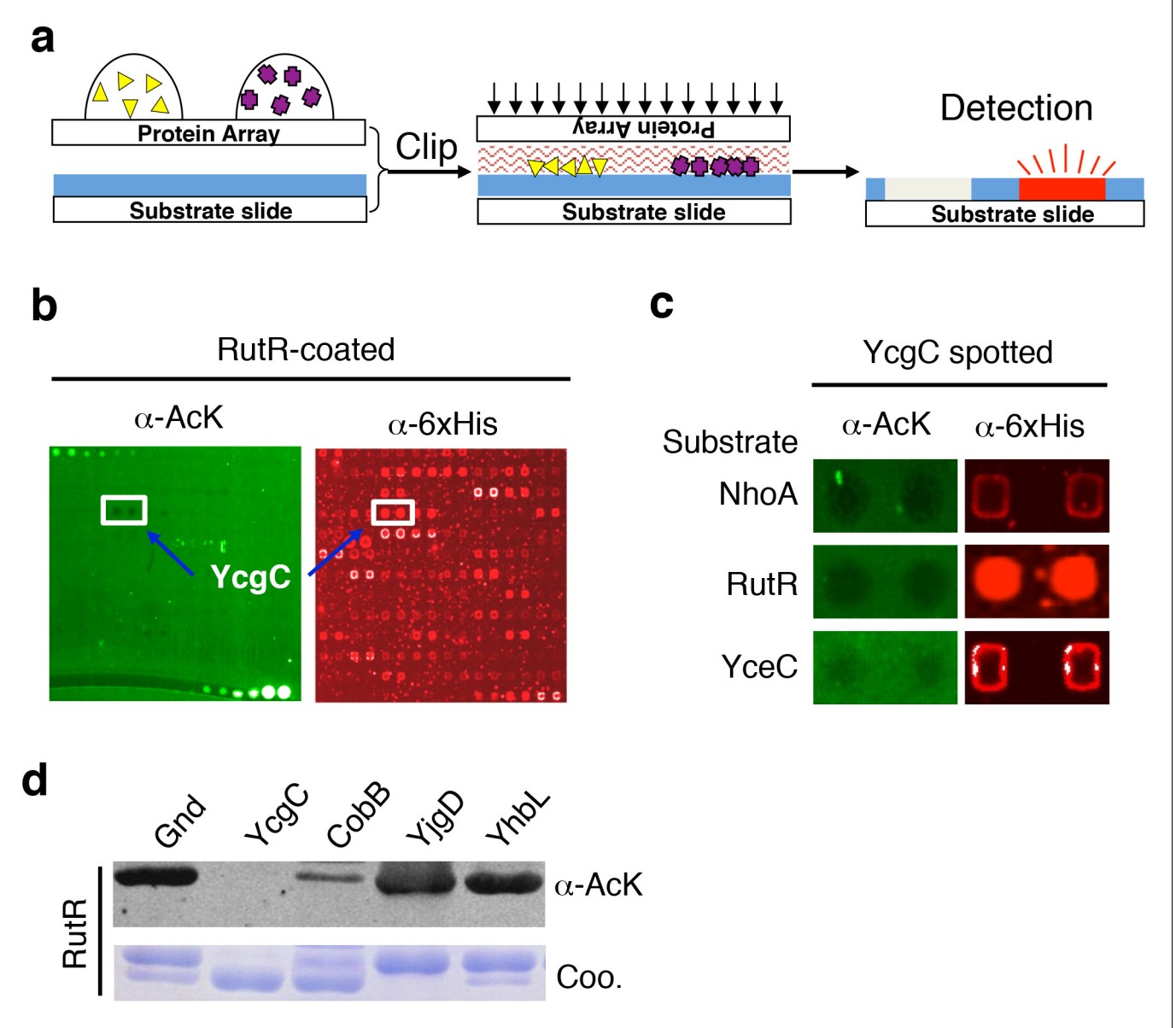

**Figure 1.** Screening the *Escherichia coli* proteome to discover new KDACs using the 'clip-chip' strategy. (a) Schematic of the 'clip-chip' strategy. (b,c) Identification of YcgC as a potential protein deacetylase. *E. coli* proteome chips were clipped onto three substrate slides separately coated with acetylated RutR, NhoA, and YceC. After incubation in a protein deacetylase buffer, the reactions were terminated by adding wash buffers, followed by a signal detection step with a pan α-AcK antibody coupled with a Cy3-labeled secondary antibody as detection reagent to visualize the loss of signals (e.g., black holes in (b,c). To determine the identity of proteins that generated the holes, the substrate slide was subsequently probed with an α-6xHis antibody followed by a Cy5-conjugated secondary antibody. (d) Using acetylated RutR proteins purified from *E. coli*, of the four candidates tested, YcgC showed robust deacetylation activity in vitro. Equal amounts of RutR proteins were used in each reaction and loss of acetylation was detected with the pan α-AcK antibody.

The following figure supplement is available for figure 1:

**Figure supplement 1.** Design of the 'clip-chip' strategy.

of black holes in fluorescence images of the substrate slides (*Figure 1b,c*). To help determine the identity of the proteins with potential KDAC activity, we subsequently probed the clipped substrate slides with an α-6xHis antibody to visualize the *E. coli* proteins delivered onto the substrate slides. As a negative control, substrate slides were also processed in parallel without the clipping step. We

identified four candidates that showed significant deacetylation activities against at least one of the three substrates tested.

To validate the KDAC activity observed above, we purified the four candidate proteins and performed solution phase deacetylation reactions against RutR. CobB was also included for comparison. By evaluating the decrease in acetylation signals using an immunoblot assay with α-AcK, we confirmed that one of the candidates, YcgC, could readily deacetylate RutR in vitro, and that CobB also deacetylated RutR. YjgD did not show any detectable deacetylation activity against RutR, while Gnd and YhbL showed slight activity (*Figure 1d*). As YcgC also showed KDAC activity against NhoA and YceC (data not shown), we then focused on characterizing the function of YcgC. YcgC is previously known as DhaM, a subunit of dihydroacetone kinase complex and a nonessential gene in *E. coli*. Because the endogenous level of YcgC is very low, YcgC was overexpressed on the wild-type background in the subsequent experiments.

## In vitro characterization of YcgC's KDAC activity

As the M subunit of the dihydroxyacetone kinase complex, the possibility that YcgC has intrinsic enzymatic activity has not been reported previously (*Molin et al., 2003*). Therefore, we first determined whether YcgC also requires $NAD^+$ and/or $Zn^{2+}$ to deacetylate RutR, as class III deacetylases require $NAD^+$ as a cofactor and other classes are dependent on $Zn^{2+}$ (*Thiagalingam et al., 2003*). We chose RutR as the substrate for YcgC, because endogenous RutR proteins are highly acetylated and because it is known to regulate genes directly or indirectly involved in the complex pathways of pyrimidine and purine metabolism (*Shimada et al., 2007*; *Shimada et al., 2008*). To reduce possible contamination from other proteins, cofactors, or metal ions as much as possible, both 6xHis-tagged YcgC and RutR proteins were affinity purified from *E. coli* under stringent wash conditions, followed by overnight dialysis. Immunoblotting of the deacetylation reactions clearly showed that YcgC could deacetylate RutR effectively, but this activity did not appear to be dependent on $NAD^+$ or $Zn^{2+}$ (*Figure 2a*, *Figure 2—figure supplement 1*). High-performance liquid chromatography analysis and inductively coupled plasma-mass spectrometry (ICP-MS), respectively, confirmed that there was no detectable $NAD^+$ or $Zn^{2+}$ in the reaction (data not shown). Of note, Coomassie staining of the decaetylated RutR protein product band appeared at a slightly lower molecular weight than acetylated RutR and this is explored below.

Next, we employed liquid chromatography–mass spectrometry (LC-MS/MS) to determine which acetylated lysine residues of RutR were deacetylated by YcgC. We found that Lys52 and Lys62, present in the peptide sequences LEQIAELAGVSK[52]TNLLYYFPSK and TNLLYYFPSK[62]EALYIAVLR, respectively, were acetylated in RutR expressed in wild-type (WT) cells (*Figure 2b* and *Figure 2—figure supplement 2a*). However, after RutR was incubated with YcgC, acetylation of K52 or K62 was no longer detectable (*Figure 2c* and *Figure 2—figure supplement 2b*). Therefore, YcgC effectively deacetylates RutR on residues K52 and K62 in vitro.

## In vivo validation of YcgC's KDAC activity

To determine whether YcgC could deacetylate RutR in cells, we performed immunoprecipitation (IP)-coupled immunoblotting to measure changes in the acetylation levels of RutR over the period of YcgC induction. To enable immunoprecipitation of endogenous RutR proteins, a 3xFLAG tag was chromosomally inserted into the 3′-end of the *rutR* coding sequence. An isopropyl-beta-D-thiogalactopyranoside (IPTG)-inducible *ycgC* construct was then transformed into the *rutR:3xFLAG* cells and induced for YcgC expression for up to 4 hr. Using IP-coupled immunoblotting analysis, we observed that acetylation levels of RutR proteins were significantly reduced in a YcgC expression level-dependent manner as detected by a custom-made α-YcgC monoclonal antibody (*Figure 2d* and *Figure 2—figure supplement 3*). In contrast, the total amount of RutR was not affected by YcgC induction (*Figure 2d*). These results confirmed that YcgC effectively deacetylates RutR in vivo without affecting its stability.

To examine whether K52 and K62 acetylation sites of RutR were deacetylated by YcgC in vivo, we created two single (K52Q; K62Q)- and one double (K52/62Q)-mutants of RutR. After transformation of these mutants into *E. coli*, subsequent IP-coupled immunoblotting demonstrated that, compared with WT RutR, mutation of either K52 or K62 resulted in a substantial loss of acetylation signals in RutR, with the K52/62Q double mutant showing the lowest acetylation signals (*Figure 2e*). RutR

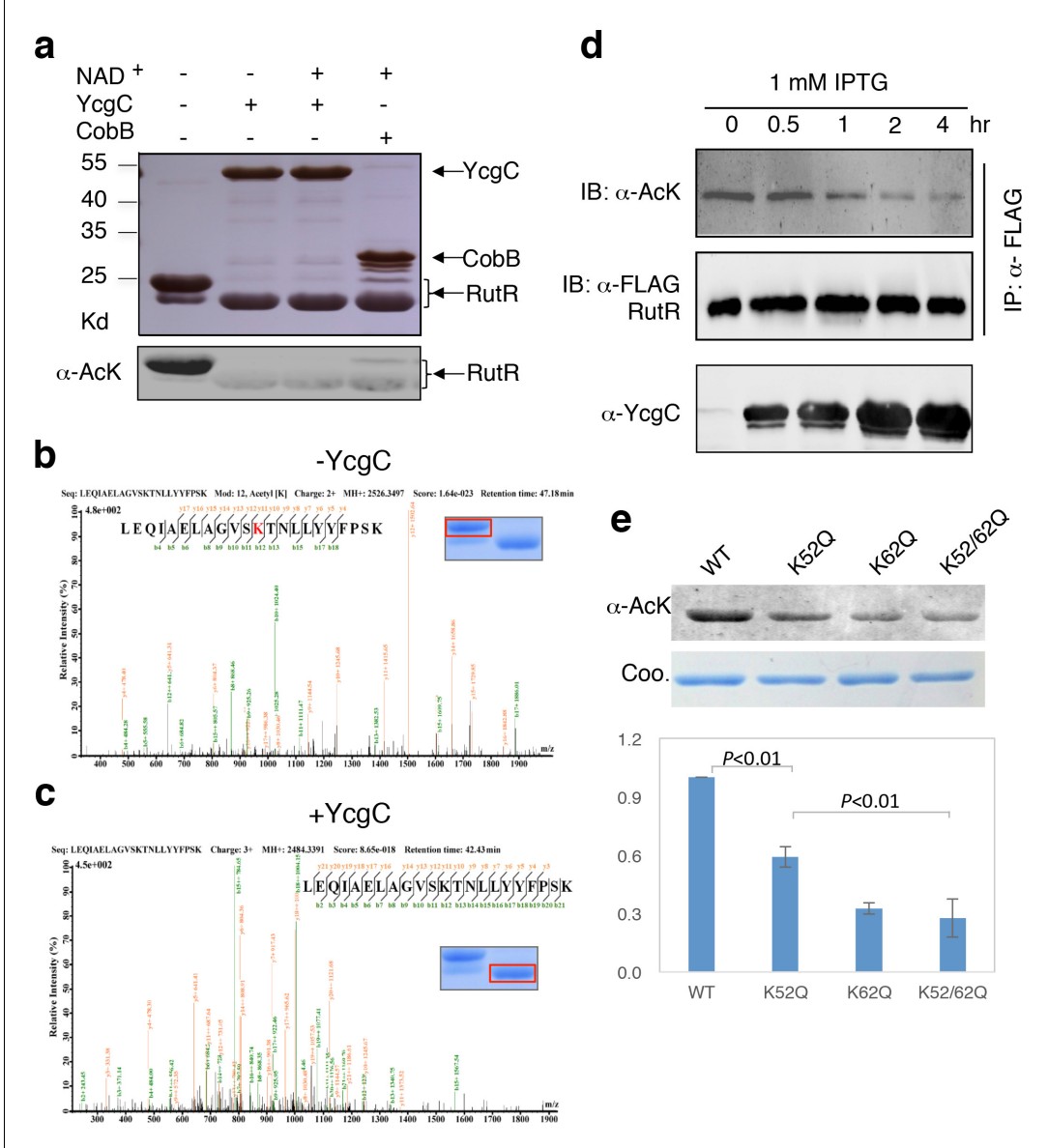

**Figure 2.** In vitro and in vivo characterization of YcgC's KDAC activity. (**a**) In vitro assays of the KDAC activity of YcgC on RutR demonstrated that its KDAC activity does not require either $NAD^+$ or $Zn^{2+}$ as cofactors. Incubation with YcgC almost completely abolished the slower migrating acetylated RutR bands (upper panel) as evidenced by immonublotting (lower panel). (**b,c**) LC-MS/MS analysis to determine the residues of RutR deacetylated by YcgC. RutR was treated with YcgC first and the untreated RutR used as the control. Both these two samples were resolved on a SDS-PAGE gel side by side. The upper band represents the Kac-containing starting materials and the lower band represents the K-containing product, which were then recovered from the gel and subjected for MS/MS analysis (inserts). Lys52 was identified as an acetylated site in RutR protein (**b**). After incubating with YcgC, acetylation on K52 was no longer detectable (**c**). (**d**) RutR is deacetylated by YcgC in *Escherichia coli*. A 3xFLAG tag was chromosomally inserted at the 3'-end of *rutR* coding sequence. Acetylation of 3xFLAG-tagged RutR was monitored upon induction of YcgC. While the protein level of RutR was unchanged (middle panel), its acetylation level was dramatically reduced as a function of YcgG induction (upper panel). YcgC's expression was monitored using a custom-made antibody (lower panel). (**e**) Mutagenesis of RutR confirmed that K52 and K62 are acetylated in vivo. Two single mutants K52Q and K62Q and one double mutant K52/62Q were constructed. These mutants along with WT RutR were produced and purified in *E. coli*. Equal amounts of purified proteins were Western blotted with the α-AcK antibody, quantitation of acetylation level of these samples were performed. KDAC: Lysine deacetylase; LC-MS/MS: Liquid chromatography–mass spectrometry; IP: Immunoprecipitation.

The following figure supplements are available for figure 2:

**Figure supplement 1.** Without $NAD^+$, CobB could not deacetylate RutR.

**Figure supplement 2.** LC-MS/MS analysis to determine the residues of RutR deacetylated by YcgC.

*Figure 2 continued on next page*

*Figure 2 continued*

**Figure supplement 3.** Specificity and sensitivity of the custom-made YcgC monoclonal antibody as assessed by Western blotting.

**Figure supplement 4.** Mutagenesis of RutR confirmed that K52 and K62 are acetylated in vivo.

K-to-R mutants (i.e., K52R, K62R, and K52/62R) were also created and tested, and similar results were observed to those with the K-to-Q mutants (*Figure 2—figure supplement 4*). These results suggest that both K52 and K62 are major acetylation sites in RutR, and can be effectively deacetylated by YcgC in *E. coli*.

## YcgC belongs to the serine hydrolase family

Because KDACs catalyze hydrolytic reactions on lysine residues, we tested a variety of hydrolase inhibitors against YcgC in in vitro deacetylation reactions as described above. We found that Halt Protease Inhibitor Cocktail (with or without ethylene glycol tetraacetic acid [EGTA]; Thermo Scientific, Rockford, IL) and Complete Protease Tablet (Roche, Mannheim, Germany) could significantly inhibit YcgC's deacetylase activity (*Figure 3—figure supplement 1a*). Further analysis revealed that the active component in the Halt Protease Inhibitor Cocktail was a serine hydrolase inhibitor 4-(2-aminoethyl)benzenesulfonyl fluoride (AEBSF), but not the other components. Additional assays demonstrated that YcgC's deacetylase activity could not be inhibited by well-known deacetylase inhibitors, including trichostatin A, SAHA (suberoylanilide hydroxamic acid), and NAM (Nicotinamide), and that hydrolase inhibitors, phenylmethylsulfonyl (PMSF), leupeptin, ethylenediaminetetracetic acid and EGTA, showed no detectable inhibition of YcgC (*Figure 3—figure supplement 1b,c*). Thus, it is likely that YcgC belongs to the serine hydrolase family, which has no significant homology to any of the annotated KDACs to date.

To identify which Ser residue in YcgC was most critical for its hydrolase (KDAC) activity, we examined five Ser residues, namely S7, S10, S73, S77, and S200, which are highly conserved among its prokaryotic homologs on the basis of protein sequence alignment. Next, we created a quintuple Ser-to-Ala mutant (i.e., 5SA) in YcgC and tested its ability to deacetylate RutR in vitro. As compared with the WT YcgC, the 5SA mutant appeared devoid of RutR deacetylase activity (*Figure 3a*). To determine which of the five Ser residues was likely to be the catalytic nucleophile for hydrolase activity, we incubated WT YcgC with hydrolase inhibitor AEBSF and the following MS/MS analysis revealed that Ser200 was the only conserved residue that was covalently labeled with AEBSF (*Figure 3b*). To confirm its importance, a Ser200-to-Ala (S200A) mutant was created and tested in the same in vitro assay. As anticipated, the deacetylase activity was not detectable with S200A YcgC, establishing Ser200 as the likely key catalytic residue.

As mentioned, RutR deacetylation by YcgC reproducibly induces faster migration on sodium dodecyl sulfate polyacrylamide gel electrophoresis (SDS-PAGE). This behavior was also observed with deacetylation by the sirtuin family member CobB (*Figure 2a*; *Figure 3a*). This phenomenon is not typically associated with protein deacetylation and we considered the possibility that proteolytic degradation of deacetylated RutR was occurring. N-terminal Edman sequencing of RutR after YcgC treatment showed N-terminal truncation with loss of 14 residues (N-MTQGAVKTTGKRSR) or 11 residues (N-MTQGAVKTTGK). RutR without YcgC treatment was also sequenced, no N-terminal truncation was observed. We considered the possibility that YcgC might show protease activity in addition to its unambiguous deacetylase activity. However, we believe this is unlikely since treatment of RutR with the sirtuin CobB, mechanistically and structurally unrelated to YcgC, induces a similar N-terminal truncation in RutR, visualized by SDS-PAGE and detected by N-terminal sequencing with loss of 14 residues. We suspect that this N-terminal cleavage of RutR results from autoproteolysis. To explore this possibility, intact RutR proteins were first heat-denatured and then incubated with either YcgC or CobB under the same deacetylation conditions. As shown in *Figure 3*, heat-denatured RutR could be partially deacetylated by both YcgC and CobB, but did not show a downshifting on SDS-PAGE. These results are consistent with the possibility that deacetylation of native acetylated RutR sparks autoproteolysis but denaturation inhibits this autoproteolytic activity.

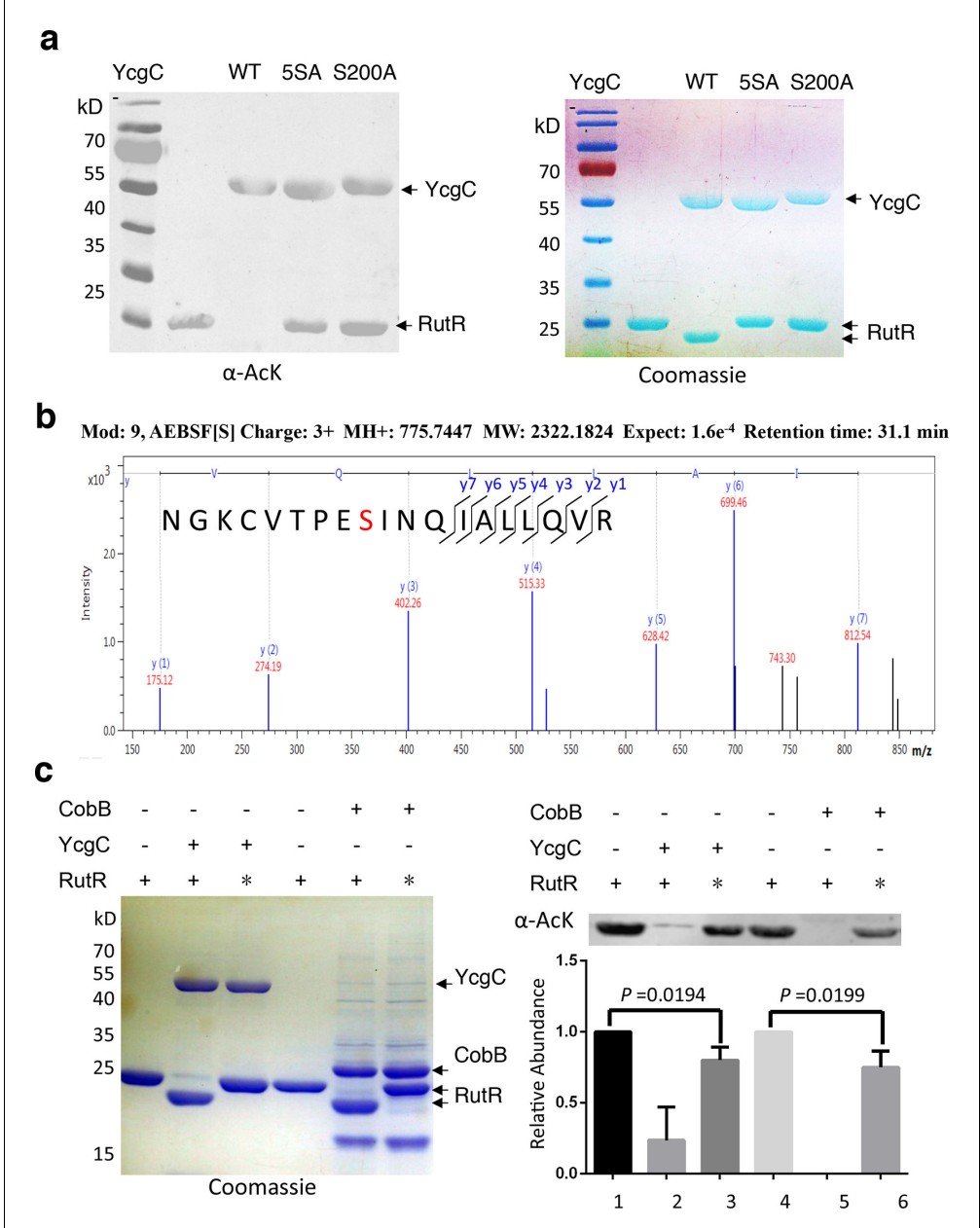

**Figure 3.** S200 is critical for YcgC's deacetylation activity. (a) Two mutants of YcgC, that is, S S8/10/73/77/200A and S200A were constructed through gene synthesis. In vitro assays of the KDAC activity of these two mutants on RutR demonstrated that their KDAC activities were completely abolished. (b) S200 on YcgC was identified as an AEBSF binding site by LC-MS/MS analysis. YcgC was incubated with AEBSF, then trypsin digested and subjected to MS/MS analysis. Upon AEBSF-mediated sulfonation, a 183 Da molecular weight increase is predicted. (c) In vitro assays of the KDAC activity of YcgC on heat-denatured RutR demonstrated that YcgC was still active (right panel), while the downshift band disappeared (left panel). Similar results were observed when heat-denatured RutR was treated with CobB. KDAC: Lysine deacetylase; LC-MS/MS: Liquid chromatography–mass spectrometry; AEBSF: 4-(2-Aminoethyl)benzenesulfonyl fluoride; WT: Wild type.

The following figure supplements are available for figure 3:

**Figure supplement 1.** YcgC's protein deacetylase activity is inhibited by AEBSF.

**Figure supplement 2.** Lysine 62 is critical for RutR proteolysis.

**Figure supplement 3.** The $K_m$ and $V_{max}$ values of YcgC were determined using RutR as a substrate.

To understand whether specific Lys residue(s) play a role in facilitating RutR autoproteolysis, K52R, K62R, K52/62R, K52Q, K62Q, and K52/62Q mutant and WT RutR protein were treated with YcgC and analyzed using Coomassie-stained SDS-PAGE (*Figure 3—figure supplement 2*). Measurement of the ratios of cleaved/intact RutR revealed that K52Q and K52R RutR behaved similar to WT RutR. In contrast, K62Q, K62R, and the two double mutant RutR proteins showed diminished cleavage. Therefore, we propose that the apparent autoproteolytic activity of RutR is dependent on its deacetylation and that removal of the acetyl group from K62 appears most important for its cleavage. Future studies will be needed to further understand the molecular mechanism for the apparent RutR autocleavage and its biological function.

Because the N-terminal cleavage of RutR is tightly coupled with its deacetylation by YcgC, the downshifted band of RutR in the YcgC deactylation reaction can be conveniently used as a surrogate of YcgC's activity. We thus performed steady-state assays to monitor the enzyme kinetics of YcgC via measuring the production of the downshifted RutR band, and estimated the $K_m$ and $V_{max}$ of YcgC to be $2.13 \pm 0.65$ μM and $0.29 \pm 0.07$ μM/min/μM, respectively (*Figure 3—figure supplement 3*).

## YcgC regulates transcription through deacetylating RutR

Originally identified as a transcriptional repressor of the *rutABCDEFG* operon in *E. coli*, RutR (YcdC) was later found to bind to 19 additional *E. coil* chromosomal loci (*Shimada et al., 2008*), including the coding regions of *pmrD* and *gcd* (*Umezawa et al., 2008*). However, deletion of *rutR* alone does not result in a significant elevation of the expression levels of most of its target genes, suggesting that RutR regulates target gene transcription via a different mechanism. To determine whether deacetylation of RutR by YcgC plays a direct role in transcription regulation of RutR's downstream target genes, we monitored the expression levels of 15 known target genes of RutR in *ycgC*-induced cells. Interestingly, while induction of *ycgC* did not change the expression level of *rutR*, the expression of two *rutR* targeting genes, *pmrD* and *gcd*, was significantly decreased, by as much as fivefold as measured with quantitative polymerase chain reaction (PCR) over a 2-hr period of *ycgC* induction (*Figure 4a*). On the other hand, induction of *cobB* in parallel did not affect expression levels of *rutR* or any of the 15 of RutR's target genes (data not shown), suggesting the possibility that YcgC regulates a different set of substrates from CobB.

To test this hypothesis, we examined the impact of induction of *ycgC* and *cobB* on global gene expression profiles in *E. coli* (*Allard et al., 1999*; *Yeung et al., 2004*). Using a standard DNA microarray approach, we found that, compared with WT cells, 197 genes were significantly repressed and 93 genes were activated after 4 hr of *ycgC* induction. In agreement with the above observations, expression levels of *pmrD* and *gcd* were significantly reduced (*Figure 4b*; *Supplementary file 1*). A similar analysis in parallel revealed that 4 hr of *cobB* induction resulted in 195 and 136 up- and down-regulated genes, respectively, compared with WT cells. However, with the exception of *xerC*, none of the RutR's targets was affected (*Supplementary file 2*). Furthermore, Venn diagram analysis did not reveal any significant overlap between either the up- or down-regulated gene groups in the *ycgC* and *cobB* induction experiments (*Figure 4b*). These results suggest that YcgC profoundly affects global gene expression and it probably functions via distinct biological processes from CobB.

This conclusion is further supported by evidence obtained at the protein level. Using immunoblotting with pan α-AcK, we observed that overexpression of *ycgC* decreases the acetylation levels of many proteins, resulting in a global change in acetylation profiles compared with those of WT cells (*Figure 4c*). Importantly, changes in the acetylation profile of *ycgC*-induced cells were different from those in *cobB*-induced cells. For example, in boxed areas 2 and 3 (*Figure 4c*), the acetylated bands are almost completely absent in *ycgC*-overexpressing cells, while they are essentially unchanged in *cobB*-overexpressing cells. On the other hand, in boxed area 1, *cobB* overexpression completely abolished the acetylation signals, whereas only a modest decrease in acetylation signals is observed in *ycgC*-overexpressing cells (*Figure 4c*). Taken together, the above results suggest that YcgC and CobB each target a distinct set of substrates.

## YcgC's homologs show protein deacetylase activities

To determine whether YcgC's KDAC activity is evolutionarily conserved, we searched for its homologs in both eukaryotes and prokaryotes. Although YcgC shows limited homology to components of the phosphotransferase system (*Punta et al., 2012*), no statistically significant homologs were

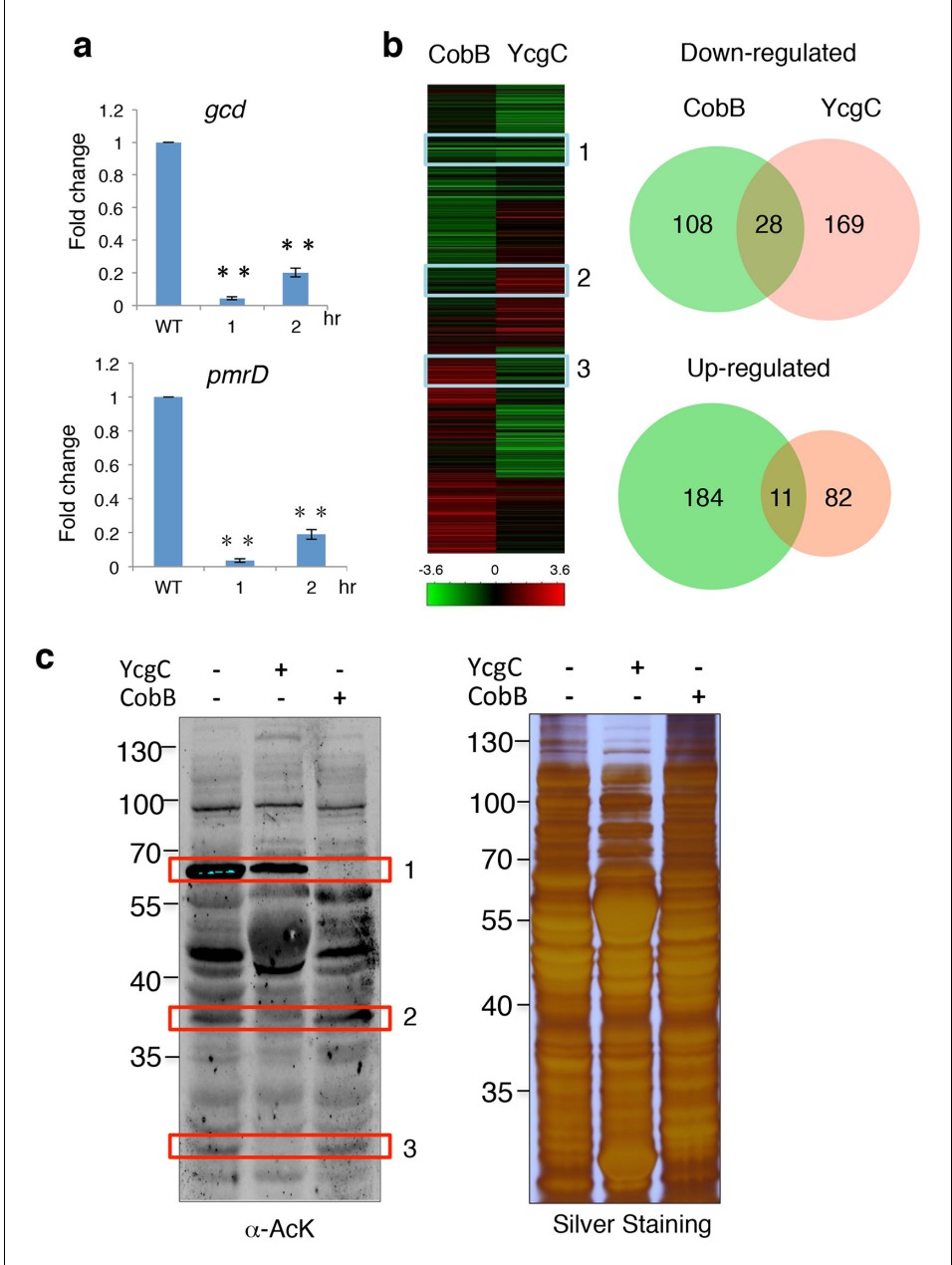

**Figure 4.** YcgC and CobB target distinct sets of substrates. (a) YcgC regulates gene expression via deacetylating RutR. Expression of *gcd* and *pmrD* is significantly reduced upon RutR induction over a period of 2 hr as measured by quantitative real-time PCR. Double asterisks indicate that the observed fold changes are statistically significant, p<0.01. (b) Global gene expression analysis of *ycgC*- and *cobB*-induced cells. Clustering analysis shows clearly that impact on global transcription of induction of *ycgC* is distinct from that of *cobB*. Venn diagram showing that there is no significant overlap between genes down- and up-regulated due to CobB and YcgC induction. (c) Overexpression of YcgC affects global protein acetylation levels in *E. coli*. After *ycgC* and *cobB* were separately induced for 1 hr, global acetylation was detected in whole lysates of *Escherichia coli* using two pan α-AcK antibodies. The WT *E. coli* strain was also included for comparison. Boxed areas indicate regions that show obviously different staining patterns in *ycgC*- and *cobB*-induced cells. PCR: Polymerase chain reaction; WT: Wild type.

identified in eukaryotes (*Molin et al., 2003*). However, many prokaryotic homologs with high similarity were readily identified. Other than homologs from *Escherichia* strains, the closet homolog is the DhaM protein from *Shigella* sp. (str. 2457T), and statistically significant homologs were also identified in more remotely related bacterial species. Therefore, we selected five homologs, representing a wide range of homology, for closer scrutiny (*Figure 5a*). Sequence alignment of the five

representative homologs with YcgC showed that the N-terminal regions (i.e., amino acids 1–2501–250) of these proteins are more conserved than the C-terminal regions (*Figure 5b*). To determine whether these YcgC homologs possess protein deacetylase activity, the five selected genes were synthesized, subcloned into the same IPTG-inducible expression vector as that of *E. coli* YcgC, and transformed into *E. coli*. After a 4 hr induction of each YcgC homolog, changes in global acetylation profiles were determined with two pan α-AcK antibodies and compared with that for WT cells (*Figure 5c,d*). Results clearly showed that induction of each of the five YcgC homologs gave rise to a unique protein deacetylation signature that is different from that of the WT and *YcgC*-induced strains. For example, overexpression of the *Klebsiella* homolog significantly reduced the acetylation levels of proteins at ~55 kDa (red box; *Figure 5c*). As another example, overexpression of the *Pantoea* homolog substantially reduced the acetylation levels of proteins around 60 and 43 kDa compared with the WT strain (wide and narrow red boxes; *Figure 5d*). Taken together, these results demonstrate that all five YcgC homologs possess readily detectable KDAC activity with different substrate preferences. Because YcgC and its homologs share little similarity with all the known KDACs identified so far, and because its activity does not require $NAD^+$ or $Zn^{2+}$, these results strongly suggest that this group represents a novel prokaryotic KDAC family.

## Discussion

In this study, we have applied a clip-chip approach to identify new KDAC candidates in *E. coli*. Our in-depth biochemical characterization revealed that the novel KDAC YcgC removes the acetyl groups on K52/62 of its substrates RutR via a previously unknown Ser hydrolase activity. A surprising observation was that, after deacetylated by either YcgC or CobB, the RutR showed a significant downshift on SDS-PAGE, suggesting possible proteolytic activity. Further biochemical analysis established that this is likely autoproteolysis of RutR that is stimulated by deacetylation of K62. Our data suggest that acetylation of RutR may enhance its stability. Indeed, endogenous RutR purified from cells grown under standard conditions is heavily acetylated. Although a complete understanding of this phenomenon will require future study; to our knowledge, this is the first example of a protein deacetylation event driving proteolytic activation.

Our in vivo functional studies on YcgC revealed that it down-regulates the expression of several RutR target genes by catalyzing the deacetylation of two lysine residues on RutR. It has been puzzling until now how RutR represses target gene transcription by its previously reported binding to coding regions of *pmrD* and *gcd*, as the binding sites are located hundreds of base pairs downstream of the start codons, and deletion of *rutR* does not enhance their expression levels (*Umezawa et al., 2008*). Based on the results of this study, we propose that YcgC deacetylates RutR leading in turn to the recruitment of additional cofactors that enhance silencing of target gene expression. As PmrD has been demonstrated to serve as a connector between multiple two-component signal-transduction systems in *Salmonella enterica, E. coli*, and other bacteria, our study also suggests the possibility of crosstalk between protein acetylation and phosphorylation in *E. coli*, a prevalent regulatory mechanism found in eukaryotes (*Eguchi and Utsumi, 2005*).

This study also highlights several advantages of the clip-chip approach. First, as an activity-based screen, this method can be readily adopted to search for many other types of enzyme activities. Second, the clip-chip approach does not require any prior knowledge of the enzymes of interest as long as a robust biochemical assay is available. Third, it is a proteome-wide, high-throughput screen that does not require further deconvolution (e.g., in MS/MS) of the positive signals because of the use of a protein microarray on which each protein is physically addressable. Finally, the clip-chip approach is capable of functional annotation of enzymes using both gain- and loss-of-signal reactions. We envision that the 'clip-chip' strategy will proved to be of wide application for the de novo discovery of enzyme activity in biology.

## Materials and methods

### Chemicals and reagents

Unless otherwise stated, all chemicals used in this study were purchased from Sigma-Aldrich (St Louis, MO), and enzymes were purchased from New England Biolabs (Ipswich, MA).

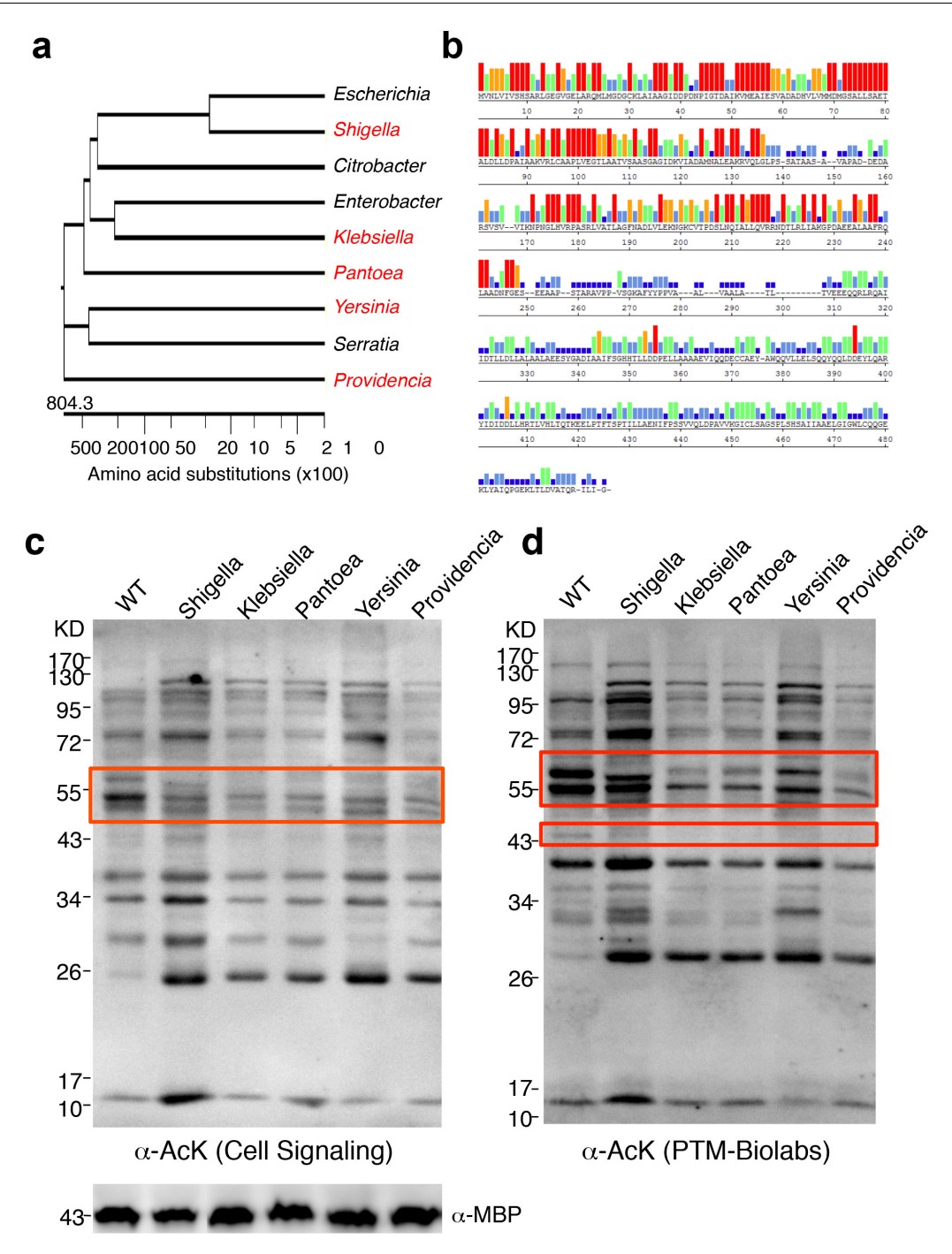

**Figure 5.** YcgC represents a new family of KDACs. (a) Five representative YcgC homologs with protein sequence homology ranging from low to high. (b) Amino acid sequence homology analysis between YcgC and five selected YcgC homologs from other bacteria. The consensus strength among the six homologous proteins at each amino acid position of YcgC is indicated with colored bars. Red, orange, green, light blue, dark blue, and blank bars represent 100, 80, 60, 40, 20, and 0% consensus strength, respectively. (c,d) Changes in global *E. coli* acetylation profiles upon induction of the five YcgC homologs. The five selected YcgC homologs were cloned, transformed into *E. coli*, and induced to overexpress. Global acetylation profiles of each induced strain were detected with a pan monoclonal antibody (Cell Signaling, #9441) and a pan polyclonal antibody (PTM-Biolabs, PTM-105), as shown in **c** and **d**, respectively. WT *E. coli* cells were also processed in parallel as a comparison. An antibody against myelin basic protein was used as a loading control. WT: Wild type.

### E. coli proteome chip preparation

*E. coli* proteome chips were prepared as described previously (*Chen et al., 2008*). In brief, expression plasmid-carrying *E. coli* cells were cultured, induced, and harvested in 96 deep-well plates. To purify the fusion proteins, cell pellets were treated with lysozyme and incubated with Ni-NTA Superflow (QIAGEN, Valencia, CA) in Multiscreen Nylon Mesh filter plates (Millipore, Billerica, MA). After six washes, the proteins were eluted with 250 mM imidazole. To prepare the proteome chip, the purified proteins were re-arrayed from 96-well plates into 384-well plates in a cold room using an Apricot system (Apricot Designs, Covina, CA). The re-arrayed proteins were printed in duplicate onto plain glass slides.

### Discovery of new protein deacetylases using an E. coli proteome chip

A substrate slide was prepared by coating FAST slides with 200 µl acetylated protein at a protein concentration ≥0.1 µg/µL. The *E. coli* proteome chip was imprinted onto the substrate slide. After the removal of the proteome chip, the substrate slide was submerged in protein deacetylation buffer (50 mM Tris-HCl, 4 mM MgCl$_2$, 50 mM NaCl, 50 mM KCl, 1 mM NAD$^+$ , pH 8.0). The reaction was carried out at 26°C for 16 hr. The slide was washed three times with 1× Tris-buffered saline and Tween 20 (TBST), 5 min each time, and incubated with an α-AcK antibody (#9441 of Cell Signaling Technology, Danvers, MA). The incubation was carried out with a 1:1000 antibody dilution at room temperature for 1 hr. The slide was washed four times with 1× TBST, 5 min each time, followed by incubation with a Cy3-conjugated secondary antibody from Jackson ImmunoResearch (West Grove, PA). To facilitate the identification of positive spots, the substrate slide was further probed with an α-6xHis antibody followed by a Cy5 conjugated secondary antibody from Jackson ImmunoResearch. A GenePix 4200A microarray scanner was used to record the results. Since this is a loss-of-signal assay, the signal intensity of each protein spot was defined as 'Background-Foreground'. The signal intensity of each protein was averaged from the two replicate spots. Signal-to-noise ratio (SNR), that is, signal/standard deviation of background, was set as the final signal of each protein. The cutoff to call protein deacetylase candidates was set as SNR ≥3.

### Protein deacetylase assays

Deacetylase candidates identified by clip-chip were overexpressed and purified in *E. coli*. In a 20 µL reaction, the three acetylated substrates, that is, 3 µg of RutR and YceC, and 0.5 µg of NhoA, were individually incubated with 5 µg of each deacetylase candidate. The reactions were carried out in protein deacetylation buffer at 37°C for 1 hr. These protein samples were then analyzed by both silver staining and Western blotting. Membranes were further probed with an IRDye 800 secondary antibody at room temperature for 1 hr and visualized with an Odyssey Infrared Imaging System from LI-COR Biosciences (Lincoln, NE).

### The deacetylation assays with hydrolase inhibitors

Hlat protease inhibitor cocktail was purchased from Thermo Scientific, cOmplete protease tablets were from Roche, AEBSF, aprotinin, bestatin, and pepstatin A were obtained from Sangon Biotech Co., Ltd (Shanghai, China). The deacetylation assays were performed as described above except for the addition of a variety of hydrolase inhibitors at appropriate concentrations. The solvents of these inhibitors, that is, dimethyl sulfoxide and ethanol, were also tested as controls.

### Measure NAD$^+$ by liquid chromatography-coupled high-resolution mass spectrometry

Affinity purified YcgC (80 µg) was diluted in 200 µL phosphate-buffered saline (pH 7.4), and denatured at 100°C for 10 min to release any bound NAD$^+$. After centrifugation at 12,000 rpm for 10 min, the supernatant was transferred to a new Eppendorf tube, and acetonitrile (ACN) was added to the supernatant at a ratio of 3:1 (vol/vol). The reaction was mixed well, allowed to stand for 20 min at 4°C, then centrifuged at 12,000 rpm for 10 min. The resulting supernatant was then subjected to liquid chromatography-coupled high-resolution mass spectrometry (LC-HRMS) analysis. An aliquot of pure NAD$^+$ (50 µM) was tested to calibrate the LC-HRMS system and as a positive control. LC-HRMS was performed as described previously (*Vogliardi et al., 2011*) on a Waters ACQUITY UPLC system equipped with a binary solvent delivery manager and a sample manager, coupled with

a Waters Micromass Q-TOF Premier Mass Spectrometer equipped with an electrospray interface (Waters Corporation, Milford, MA). Briefly, LC was performed on a Syncronis HILIC column (50 × 2.1 mm, 1.7 μm) (Thermo Scientific). The column was eluted with 200 mM ammonium formate aqueous solution and ACN in gradient mode at a flow rate of 0.30 mL/min at 30°C. MS was performed using negative polarity, 2.4 KV capillary voltage, 30 V sampling cone, 4 eV collision energy, a source temperature of 110°C, and a desolvation temperature of 350°C. The flow rate for the desolvation gas was set at 600 L/hr. Scan range was set to m/z 50–1000, scan time to 0.3 s, and interscan time to 0.02 s.

## Metal analysis by ICP-MS

ICP-MS analysis (*Goullé et al., 2005*) was performed according to the manufacturer's instructions. Briefly, 0.48 mg of YcgC was prepared in 4 mL concentrated nitric acid and 2 mL deionized water. The solution was then subjected to microwave digestion with a Multiwave 3000 instrument from Anton Paar ShapeTec GmbH (Wundschuh, Austria) at 600 W power for 15 min. The digested sample was filtered through a filter paper. The sample was analyzed on an ELAN 9000 ICP-MS instrument from PerkinElmer, Inc. (Waltham, MA). Human hair GBW07601a (GSH-1a) from Institute of Geophysical and Geochemical Exploration (Hebei, China) was included as a positive control. All experiments were carried out at room temperature in a dust-free area with a relative humidity of 10–85%.

## Measuring the $K_m$ and $V_{max}$ of YcgC

YcgC (0.3 μM) was incubated with RutR at a series of concentrations, that is, 1.5, 2.3, 3, 4.5, 6, 7.5, 9, 12, and 15 μM. The reaction was carried out in protein deacetylation buffer without NAD$^+$ at 37°C for 15 min. Protein samples were then resolved by 12% SDS-PAGE followed by silver staining. The gel was scanned with a PowerLook 2100XL from Techville, Inc. (Dallas, TX), converted to an 8-bit grayscale image and analyzed by Image J (NIH; http://rsb.info.nih.gov/ij/).

## Identification of deacetylation sites by MS

Acetylated RutR and deacetylated RutR were trypsin-digested and analyzed with a nanoflow LC-MS/MS coupled online with a Q Exactive Plus quadrupole orbitrap mass spectrometer (Thermo Scientific, San Jose, CA) equipped with a nanoelectrospray ion source. Briefly, the peptide mixtures were loaded onto a C18 column (100 mm inner diameter, 10 cm long, 5 mm resin) from Michrom Bioresources (Auburn, CA) using an autosampler. Peptides were eluted with a 0–35% gradient (Buffer A, 0.1% formic acid, and 5% ACN; Buffer B, 0.1% formic acid, and 95% ACN) over 80 min and detected online with a Q Exactive Plus quadrupole orbitrap mass spectrometer using a data-dependent TOP10 method (*Haas et al., 2006*).

## Construction of an *E. coli* strain (W3110) harboring chromosomal 3xFLAG-tagged RutR

*E. coli* strain (W3110) harboring chromosomal 3xFLAG-tagged RutR was constructed using the Red recombination system (*Poteete, 2001*). In short, the DNA cassette for recombination was composed of a 150 bp upstream flanking sequence, the *rutR* gene, a 3xFLAG tag before the stop codon of *rutR*, followed by the sequence of the kanamycin resistance gene, and a 150 bp downstream flanking sequence. This cassette was synthesized and cloned into pUC57 by GenScript (Nanjing, China). The cassette was amplified by high fidelity PCR and treated with DpnI. One microgram of the linear DNA fragment was electrotransported into *E. coli* W3110 cells carrying pKD46, and these recombinants were selected using kanamycin medium and verified by colony PCR.

## Determination of the deacetylation activity of YcgC in vivo

The plasmid carries ycgC from the *E. coli* AG1 strain that we used for the construction of the *E. coli* proteome chip, was extracted and transformed into the *E. coli* W3110 strain harboring chromosomal 3xFLAG-tagged RutR. The transformed *E. coli* strain was then cultured in lysogeny broth media to a OD$_{600}$ of 0.6–0.8 and induced by 1 mM IPTG at 37°C for 0.5, 1, 2, and 4 hr. Cells were harvested and treated with lysis buffer (50 mM NaH$_2$PO$_4$, 300 mM NaCl, 20 mM imidazole, 1× CelLytic B, 50 units/mL of Benzonase proteinase inhibitor cocktail, and 1 mM PMSF, pH 8.0) at 4°C for 2 hr with vigorous shaking. The 3xFLAG-tagged RutR was then immunoprecipitated using an α-FLAG

antibody and protein G conjugated agarose beads. Samples were resolved on a 10% SDS-PAGE gel followed by Western blotting with an α-AcK antibody (Cell Signaling Technology, Shanghai, China) and an α-FLAG polyclonal antibody.

### Determination of the acetylation level of RutR mutants

RutR mutants K52Q, K62Q, and double mutant K52Q/K62Q were synthesized and cloned into pET28a + with GenScript (*Supplementary file 3*). The acetylation level of these RutR mutants was detected with an α-AcK antibody (Cell Signaling Technology) and compared with that of the WT *E. coli* strain.

### Preparation of α-YcgC antibody

Mouse α-YcgC monoclonal antibody was custom-made by Abmart, Inc. (Shanghai, China). Western blotting was applied to characterize the antibody. The sensitivity of the antibody was tested using serially diluted RutR and its specificity was tested by spiking purified RutR into a whole lysate of *E. coli*.

### Identify AEBSF binding site on YcgC by LC-MS/MS

At a molar ratio of 500:1, AEBSF and YcgC were incubated at 37°C for 1 hr. After SDS-PAGE and Coomassie staining, YcgC band was cut off and in-gel trypsin digestion was done according to the standard protocol. The digested YcgC was then analyzed using Ultimate 3000 Nano Pump LC system from Thermo Scientific coupled with an electrospray ionization quadrupole time-of-flight mass spectrometer from Bruker Daltonics (Bremen, Germany). The LC setup was coupled online to a Q-TOF using a nano-ESI source from Bruker Daltonics in data-dependent acquisition mode (m/z 350–1500). Tandem mass spectra were extracted, charge state was deconvoluted and deisotoped by Compass Data Analysis version 4.1 from Bruker Daltonics. Mascot version 2.4 from Matrix Science (Boston, MA) was set up to search the database (entries). Carbamidomethyl on cysteine was specified as fixed modifications, oxidation of methionine was specified as variable modifications.

### Identification of the N-terminal sequence of deacetylated YcdC by YcgC and CobB

RutR was incubated with YcgC in deacetylation buffer at 37°C for 1 hr. Protein sample was then resolved by 12% SDS-PAGE and transferred to the polyvinyl difluoride membrane. The shift RtuR band was dyed by Ponceau S and cut off, and then N-terminal sequenced by protein sequencer PPSQ-33A from Shimadzu (Kyoto, Japan). The raw data and graphs generated by PPSQ-33A were identified and exported by PPSQ-33A data processing. The N-terminal sequence RtuR was then determined.

### Determination of the acetylation status of *E. coli* whole lysates

*E. coli* cells were cultured in LB medium at 37°C. Before and after induction with 1 mM IPTG for 1 hr, cells were treated with lysis buffer at 4°C for 2 hr with vigorous shaking. Cell debris was removed by centrifugation at 4°C. The protein concentration of the whole lysate was determined using the BCA Protein Assay (Pierce, Rockford, IL); 100 μg of whole lysate was then resolved on a 10% SDS-PAGE gel followed by Western blotting with an α-AcK antibody (Cell Signaling Technology) overnight at 4°C. As a loading control, the protein lysates were Western blotted with an α-myelin basic protein antibody from Abmart, Inc. (*Noinaj et al., 2013*; *Spanò et al., 2011*). Results were recorded using an IRDye 800 secondary antibody and the Odyssey Infrared Imaging System (LI-COR Biosciences).

### Quantitative real-time PCR

Total RNA was extracted using an RNA extraction kit from TIANGEN Biotech Co., Ltd. (Beijing, China). RNA was then reverse-transcribed to cDNA using a random oligo primer from Promega (Beijing, China), according to the manufacturer's instructions. Primers were synthesized by Sangon Biotech. (Shanghai, China) (*Supplementary file 4*) and validated by regular PCR and melting curve analysis. Real-time PCRs were carried out using FastStart Universal SYBR Green Master from Roche (Shanghai, China) and the ABI 7500 real-time PCR platform (Life Technologies Corporation, Shanghai, China).

## Global gene expression analysis using DNA chips

*E. coli* DNA chips were purchased from CapitalBio Corp. (Beijing, China). cDNA labeled with a fluorescent dye (Cy5 and Cy3-dCTP) was produced by Eberwine's linear RNA amplification method and subsequent enzymatic reaction (*Guo et al., 2005*; *Patterson et al., 2006*). Arrays were hybridized in a CapitalBio BioMixer II Hybridization Station overnight and scanned with a LuxScan scanner and the images obtained were then analyzed using LuxScan 3.0 software from CapitalBio Corp. A space- and intensity-dependent normalization based on a LOWESS program was employed (*Yang et al., 2002*). To identify significantly differentially expressed genes, SAM 3.02 was used. Unsupervised hierarchical clustering was used to cluster samples or genes. The distance between single samples or genes was based on Pearson's correlation coefficients. Distances between clusters were calculated using the 'complete linkage' method. Venn diagrams were drawn using the R package *Vennerable.* The size of each circle proportionally reflects the number of unique genes in each group.

## Homology analysis, phylogenetic tree construction, and sequence alignment

To identify YcgC's bacterial homologs, 'dihydroxyacetone kinase subunit M' was used as a search term in PubMed under the 'Protein' category. This search showed that the top taxonomic groups, that is, *Escherichia, Klebsiella, Shigella, Serratia, Citrobacter, Yersinia, Enterobacter, Salmonella, Pantoea,* and *Providencia*, all belonged to the Enterobacteriaceae. The amino acid sequence of YcgC was then Blasted against the most significant taxonomic groups using BlastP. Bacterial strains of the closest homologs from each taxonomic group were determined, that is, *Citrobacter koseri ATCC BAA-895, Enterobacter aerogenes KCTC 2190, Klebsiella oxytoca KCTC 1686, Pantoea ananatis LMG 5342, Providencia stuartii MRSN 2154, Serratia odorifera 4Rx13, Shigella flexneri 2a str. 2457T,* and *Yersinia enterocolitica subsp. enterocolitica WA-314*, and the amino acid sequences of these homologs, along with that for YcgC from *E. coli K12 W3110*, were then subjected to phylogenetic tree construction and sequence alignment using LaserGene (DNAstar Inc. Madison, WI).

## Acknowledgements

The authors are grateful to W Yan, YK Wan, PY Yang, and MJ Tan for their expert research assistance and comments and J Fleming for editing the manuscript. This study was supported in part by grants from the National Natural Science Foundation of China (No. 31370813 and 31000388), and the National High Technology Research and Development Program of China (No. 2012AA020103 and 2012AA020203) to SCT, grant from National Natural Science Foundation of China (No. 31370750) to DMC, grants from the NIH (RR020839, GM076102, and HG006434) to HZ, and grant NIH GM62437 to PAC. We thank the FAMRI (Flight Attendant Medical Research Institute) Foundation for support of this work.

## Additional information

### Competing interests

PAC: Reviewing editor, *eLife*. The other authors declare that no competing interests exist.

### Funding

| Funder | Grant reference number | Author |
|---|---|---|
| National Natural Science Foundation of China | 31370750 | Daniel M. Czajkowsky |
| National Institutes of Health | GM62437 | Philip A Cole |
| Flight Attendant Medical Research Institute | | Philip A Cole |
| National Institutes of Health | RR020839 | Heng Zhu |
| National Institutes of Health | GM076102 | Heng Zhu |
| National Institutes of Health | HG006434 | Heng Zhu |

| | | |
|---|---|---|
| Ministry of Science and Technology of the People's Republic of China | 2010CB529205 | Sheng-Ce Tao |
| National Natural Science Foundation of China | 31370813 | Sheng-Ce Tao |
| National Natural Science Foundation of China | 31000388 | Sheng-Ce Tao |
| Ministry of Science and Technology of the People's Republic of China | 2012AA020103 | Sheng-Ce Tao |
| Ministry of Science and Technology of the People's Republic of China | 2012AA020203 | Sheng-Ce Tao |
| Ministry of Health of the People's Republic of China | 2013ZX10003006 | Sheng-Ce Tao |

The funders had no role in study design, data collection and interpretation, or the decision to submit the work for publication.

## Author contributions
ST, S-JG, C-SC, C-XL, H-WJ, FG, J-YD, YL, B-RQ, Performed the experiments, Acquisition of data, Analysis and interpretation of data, Drafting or revising the article; Y-MZ, DMC, Analyzed the micro-array data, Acquisition of data, Analysis and interpretation of data, Drafting or revising the article; Y-HA, Acquisition of data, Analysis and interpretation of data, Drafting or revising the article; PAC, HZ, S-CT, Conceived and designed the study with assistance from YHA, Analyzed the data, Wrote the manuscript, Conception and design, Analysis and interpretation of data, Drafting or revising the article

# Additional files

### Supplementary files
• Supplementary file 1. Genes differentially expressed when YcgC was overexpressed.

• Supplementary file 2. Genes differentially expressed when CobB was induced.

• Supplementary file 3. Clones constructed in this study.

• Supplementary file 4. Primers used in this study.

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
