## [Decision Letter]

Thank you for sending your work entitled "YcgC Represents a New Protein Deacetylase Family in Prokaryotes" for consideration at *eLife*. Your article has been favorably evaluated by Michael Marletta (Senior Editor), Leemor Joshua-Tor (Reviewing Editor), and three reviewers.

The Reviewing Editor and the reviewers discussed their comments before we reached this decision, and the Reviewing Editor has assembled the following comments to help you prepare a revised submission.

The paper describes an interesting study that uses an application of proteome array "Clip-Chip" technology to the discovery of new enzymatic functions. This appears to be a new twist on the use of proteome arrays and therefore very appealing. This technology has clearly identified a new class of protein deacetylases that does not require metal ions or NAD^+^. This is a fortunate happenstance and highlights the utility of the method for discovery of enzymes that informatics would not have picked up by homology searching. Thus, this paper will be of considerable interest to a broad spectrum of biological scientists.

However, several issues have been raised specifically regarding the activity of YcgC as a lysine deacetylase that would have to be addressed:

1) The biochemical data supporting that YcgC has deacetylase activity was based on Western blots using pan-specific acetyl lysine antibodies. One particular concern is that YcgC might be a protease that can cleave RutR at a specific sequence. This cleavage may in turn affect the recognition of an acetyl lysine residue in RutR by the antibody, leading to a decreased acetyl lysine signal on Western blots. This is highly likely because Figure 2 shows that RutR became smaller after treatment with YcgC. In addition, further post-translational modifications adjacent to the acetylated lysine might alter the epitopes for proper readout by an antibody. To rule out this concern, more biochemical data is needed. One suggestion is to examine whether YcgC can deacetylate a synthetic acetyl peptide and monitoring products using LC-MS based methods, which can provide definitive proof for the deacetylase activity.

2) In Figure 1 legend, you state: "using acetylated RutR proteins […] YcgC showed robust deacetylation activity in vitro". It seems that the SDS-PAGE mobility of RutR doesn't change following deacetylation. However, in Figure 2, there is at least 5KD shift to a lower molecule weight after deacetylation. Such a discrepancy should be addressed.

3) The mass spectrometry data showing the acetylated K52 peptide (Figure 2) is problematic. Although the spectra itself is not readable, it is clear that the peptide was obtained by trypsin digestion (as stated in the experimental section), which cleaves after lysine and arginine, but not after acetyl lysine. A long peptide with Kac embedded in the center should be detected. Figure 2 shows a peptide cut after acetyl lysine, which is highly unlikely. This indicates that either the acetyl group is not on K52 or the MS data is unreliable. Although all the Y ions are marked, for the C-Kac peptide as shown, the Y ions shouldn't be visible because it lost the positively charged Lys which shouldn't fly in MS. In contrast, the C-Lys peptide should fly. However, the similar ionization efficiency for the b and y ions for the two peptides doesn't make sense. The images are at low resolution and therefore hard to examine in detail. In addition, HPLC retention times should be indicated. The same concern is noted for the MS data shown in the supplementary figures.

4) The gel shown in Figure 2 is not convincing. The authors state in the text that the double mutant provides the lowest level of acetylation of RutR, the substrate protein. This is not apparent. At least by my eye the K62Q and the double mutant appear to provide quite similar results. Some sort of quantification of these results should be provided.

5) It was not clear in the manuscript whether or not YcgC is an essential protein in *E. coli*. A related point is that when the authors overexpress the protein in vivo, it was not clear whether this was being done in a ∆ycgC strain or on top of the native level of YcgC. Please provide this information.

Other issues to address:

6) It is surprising that 1x or 2x-acetylation/deacetylation can cause such a dramatic mobility shift (Figure 2). Fortunately, the authors showed that CobB also deacetylates the same substrate. This data should be included to show that deacetylation can indeed cause a 5kD mobility shift.

7) In Figure 2, why are the band intensities of alpha-Flag and α-YcgC so different? In addition, even given that α-Flag bands have equal intensity, the decreased α-Kac bands should correspond to two α-Flag bands given the decreased mobility shift following deacetylation.

8) For the MS analyses, it is not unclear whether the authors run the SDS page gels first and then cut the band for MS analysis. In theory, there should be two bands (Kac-containing starting materials and K-containing products), which should be resolved well and analyzed in parallel.

9) In Figure 2, interestingly, although the negative signals of α-Kac antibodies, the Coo. bands showed the similar mobility as shown in Figure 2. It seems that the same reaction occurs when RutR was mixed with YcgC. It is another strange observation.

[Editors' note: further revisions were requested prior to acceptance, as described below.]

Thank you for resubmitting your work entitled "YcgC Represents a New Protein Deacetylase Family in Prokaryotes" for further consideration at eLife. Your revised article has been favorably evaluated by Michael Marletta (Senior Editor), a Reviewing Editor, and two reviewers. The manuscript has been significantly improved but there are some remaining issues that need to be addressed before acceptance, as outlined below:

The reviewers all felt that this work is significant and important. Precisely because of this, it is important to make sure as much as possible that observations reported are not due to a contaminating protease activity. The MS data showing 95% sequence coverage of RutR sequence cannot prove that there is no proteolysis. The fact that YcgC does not work on a synthetic peptide is another reason for concern.

There are a couple of simple things that could be done to address this concern:

1) In Figure 2, include a negative control with CobB but without NAD^+.^If there is a protease contamination, it is likely an endogenous *E. coli* protein that is present in both the YcgC and CobB prep. Using CobB without NAD^+^ should resolve this.

2) In Figure 2, K52Q and K62Q mutants were blotted and they appear to be of similar size to the WT protein. One can argue that the K to Q mutant behaves similarly to the acetylated WT protein. However, the authors should repeat these experiments using the K52R and K62R mutants. If these mutants migrate faster as the deacetylated RutR, it will also help address these concerns. In addition, the K to R mutants may improve the Ac Western blot because Q is generally thought to mimic acetyl lysine.

It is interesting that the 5-kd mobility shift only occurs for native YcgC but not N-terminal flagged YcgC. Such an observation should be described explicitly in the text to avoid potential confusion, especially if the K52R and K62R mutants don't migrate as fast. This type of shift by 1 or 2 Ac groups is quite large.

3) Another possibility is to mutate the conserved Ser residues in YcgC and show that the mutant loses deacetylase activity. There are only five conserved Ser residues and mutating them would not be a huge effort.

4) For Figure 1, the experimental details provided in the main text and response letter are still not clear. Did the authors run "Coo. Stain" on the samples and then use the same sample for the deacelylation reaction or did they run "Coo. Stain" and deacelylation with equal amounts of the two aliquots? If the latter is the case, the samples after the deacelylation reaction should be subsequently stain by a Western-compatible dye followed by anti-acetyllysine antibody.

---

## [Author Response]

*[…] However, several issues have been raised specifically regarding the activity of YcgC as a lysine deacetylase that would have to be addressed: 1) The biochemical data supporting that YcgC has deacetylase activity was based on Western blots using pan-specific acetyl lysine antibodies. One particular concern is that YcgC might be a protease that can cleave RutR at a specific sequence. This cleavage may in turn affect the recognition of an acetyl lysine residue in RutR by the antibody, leading to a decreased acetyl lysine signal on Western blots. This is highly likely because Figure 2 shows that RutR became smaller after treatment with YcgC. In addition, further post-translational modifications adjacent to the acetylated lysine might alter the epitopes for proper readout by an antibody. To rule out this concern, more biochemical data is needed. One suggestion is to examine whether YcgC can deacetylate a synthetic acetyl peptide and monitoring products using LC-MS based methods, which can provide definitive proof for the deacetylase activity.*

We thank the reviewers for giving us this opportunity to further clarify this concern. In this study, the acetylated lysine residues were determined by comparing the MS/MS peaks obtained from the upper and lower (i.e., down-shifted) RutR protein bands after PAGE separation. To make this point clearer, we have now added an insert to Figure 2, as well as to Figure 2—figure supplement 3. To demonstrate that the down-shifted RutR was *not* caused by protease activity of YcgC, we repeated the MS/MS analysis several more times against the down-shifted RutR bands recovered from PAGE gels. (Please also see our response to Point 3 below.) To our satisfaction, we obtained a peptide-coverage for up to 95% and with both the N- and C- terminal of the full-length RutR, indicating that the down-shifted RutR was very unlikely to be caused by protease cleavage. Moreover, we observed that CobB-deacetylated RutR protein also appeared as a similar down-shifted band on SDS-PAGE, providing further confirmation of our interpretation of the data. Taken together, these additional results help establish that YcgC is a bona fide deacetylase and not a protease. We have now clarified this in the second paragraph of the subsection “In vitro characterization of YcgC's KDAC activity”, and added the CobB treated RutR to Figure 2.

In addition, we have also treated RutR-derived synthetic peptides in acetylated and unacetylated forms with YcgC as recommended by the reviewers. There was no evidence that the RutR acetylated peptide was deacetylated or otherwise modified by YcgC treatment as analyzed by HPLC. Our interpretation of these results is that a folded substrate conformation or long-range recognition elements within RutR are critical for YcgC processing.

*2) In Figure 1 legend, you state: "using acetylated RutR proteins […] YcgC showed robust deacetylation activity in vitro". It seems that the SDS-PAGE mobility of RutR doesn't change following deacetylation. However, in Figure 2, there is at least 5KD shift to a lower molecule weight after deacetylation. Such a discrepancy should be addressed.*

We thank the reviewers for pointing out this “discrepancy.” The purpose of Coomassie staining in the lower panel of Figure 1 is to show that an equal amount of RutR proteins was used for each reaction. Since they were not deacetylated by YcgC, no down-shift would be expected and none as observed. We have now clarified this in the Figure 1 legend.

*3) The mass spectrometry data showing the acetylated K52 peptide (Figure 2) is problematic. Although the spectra itself is not readable, it is clear that the peptide was obtained by trypsin digestion (as stated in the experimental section), which cleaves after lysine and arginine, but not after acetyl lysine. A long peptide with Kac embedded in the center should be detected. Figure 2 shows a peptide cut after acetyl lysine, which is highly unlikely. This indicates that either the acetyl group is not on K52 or the MS data is unreliable. Although all the Y ions are marked, for the C-Kac peptide as shown, the Y ions shouldn't be visible because it lost the positively charged Lys which shouldn't fly in MS. In contrast, the C-Lys peptide should fly. However, the similar ionization efficiency for the b and y ions for the two peptides doesn't make sense. The images are at low resolution and therefore hard to examine in detail. In addition, HPLC retention times should be indicated. The same concern is noted for the MS data shown in the supplementary figures.*

We have consulted several mass spectrometry experts, and re-performed the MS/MS analysis with a different MS spectrometer (Q Exactive^TM^ Plus quadrupole orbitrap mass spectrometer) of a much higher mass resolution. Long peptides with Kac embedded in the center for both K52 and K62 were clearly observed for untreated RutR (-YcgC), while not for treated RutR (+YcgC). We have now updated Figure 2 and Figure 2—figure supplement 3 with the new MS/MS data, and provided clear images of HPLC retention times. We have also modified the manuscript accordingly (in the second paragraph of the subsection “In vivo validation of YcgC's KDAC activity”, and in the subsection “Identification of deacetylation sites by mass spectrometry").

*4) The gel shown in Figure 2 is not convincing. The authors state in the text that the double mutant provides the lowest level of acetylation of RutR, the substrate protein. This is not apparent. At least by my eye the K62Q and the double mutant appear to provide quite similar results. Some sort of quantification of these results should be provided.*

To clearly demonstrate the difference of acetylation level between the wild-type and mutant RutR proteins, we further optimized the amount of proteins and repeated the experiments for several times. We now provide quantification and statistic analyses to demonstrate the significant differences of their acetylation levels in the lower panel of Figure 2. The figure legend and main text have been updated accordingly.

*5) It was not clear in the manuscript whether or not YcgC is an essential protein in* E. coli*. A related point is that when the authors overexpress the protein in vivo, it was not clear whether this was being done in a ∆ycgC strain or on top of the native level of YcgC. Please provide this information.*

YcgC is also known as DhaM, a subunit of the dihydroacetone kinase complex. It is not essential in *E. coli*. Because the endogenous level of YcgC is very low, YcgC was overexpressed on the wild type background. We have now clarified this point in the last paragraph of the subsection “Screen new KDAC using the *E. coli* proteome microarray”.

*Other issues to address: 6) It is surprising that 1x or 2x-acetylation/deacetylation can cause such a dramatic mobility shift (Figure 2). Fortunately, the authors showed that CobB also deacetylates the same substrate. This data should be included to show that deacetylation can indeed cause a 5kD mobility shift.*

We thank the reviewers for pointing this out. We have now added the CobB reaction to Figure 2, and modified the manuscript accordingly (subsection “In vitro characterization of YcgC's KDAC activity”).

*7) In Figure 2, why are the band intensities of alpha-Flag and α-YcgC so different? In addition, even given that α-Flag bands have equal intensity, the decreased α-Kac bands should correspond to two α-Flag bands given the decreased mobility shift following deacetylation.*

In Figure 2 the anti-FLAG antibodies were used to detect the expression level of endogenous RutR, because we chromosomally tagged *RutR* gene with FLAG (subsection “In vivo validation of YcgC's KDAC activity”, first paragraph), whereas anti-YcgC mAb (created during this study) was for detecting YcgC during a time course of induction. It is evident that the endogenous level of YcgC is very low as shown in the control lane in Figure 2. The down-shifted RutR bands were only observed in our in vitro deacetylation reactions, when RutR was almost completely decetylated on both K52 and K62 (as shown by MS/MS analysis). While the evidence of Figure 2 indicates that YcgC can deacetylate endogenous RutR in vivo, it is likely to be far from complete compared with in vitro assays (Figure 2), potentially accounting for the lack of a shift. The FLAG tag may also affect the mobility, as well as other PTMs.

*8) For the MS analyses, it is not unclear whether the authors run the SDS page gels first and then cut the band for MS analysis. In theory, there should be two bands (Kac-containing starting materials and K-containing products), which should be resolved well and analyzed in parallel.*

Yes, we treated RutR with YcgC first and then ran it together with untreated RutR on a PAGE gel. After Coomassie staining, we recovered both the upper band in the untreated RutR lane and the down-shifted band in the YcgC-treated lane. Both recovered proteins were then subjected to MS analysis. Please also see our response to Point 1 above. We have now added an insert to Figure 2 and Figure 2—figure supplement 3, and have clarified this in the manuscript (Figure 2 legend; Figure 2—figure supplement 3 legend).

*9) In Figure 2, interestingly, although the negative signals of α-Kac antibodies, the Coo. bands showed the similar mobility as shown in Figure 2. It seems that the same reaction occurs when RutR was mixed with YcgC. It is another strange observation.*

In Figure 2, lysine-to-glutamine mutations were used to demonstrate that K52 and K62 were the two dominant acetylated residues. Because K-to-Q mutation has been commonly used to generate an acetylated lysine mimic due to chemical similarity of an acetamide and the Gln side chain (e.g., PMID: 23904479; 19303850; 21906795), such mutation may not generate enough structural change of the protein to cause a down-shifting of the band. To better demonstrate the differences in their acetylation level, we have repeated this assay and added statistical analyses to Figure 2. Please also see our response to Point 4 above. We have now updated Figure 2 and modified the manuscript accordingly (Figure 2 legend).

[Editors' note: further revisions were requested prior to acceptance, as described below.]

*The reviewers all felt that this work is significant and important. Precisely because of this, it is important to make sure as much as possible that observations reported are not due to a contaminating protease activity. The MS data showing 95% sequence coverage of RutR sequence cannot prove that there is no proteolysis. The fact that YcgC does not work on a synthetic peptide is another reason for concern. There are a couple of simple things that could be done to address this concern: 1) In Figure 2, include a negative control with CobB but without NAD^+.^If there is a protease contamination, it is likely an endogenous E. coli protein that is present in both the YcgC and CobB prep. Using CobB without NAD^+^ should resolve this.*

We have now included the negative control with CobB but without NAD^+^. As expected, no deacetylase activity was observed for this reaction (Figure 2—figure supplement 1), suggesting that it was unlikely due to contamination by an endogenous protease. We have added this result and modified the manuscript accordingly (subsection “In vitro characterization of YcgC's KDAC activity”, first paragraph).

*2) In Figure 2, K52Q and K62Q mutants were blotted and they appear to be of similar size to the WT protein. One can argue that the K to Q mutant behaves similarly to the acetylated WT protein. However, the authors should repeat these experiments using the K52R and K62R mutants. If these mutants migrate faster as the deacetylated RutR, it will also help address these concerns. In addition, the K to R mutants may improve the Ac Western blot because Q is generally thought to mimic acetyl lysine.*

*It is interesting that the 5-kd mobility shift only occurs for native YcgC but not N-terminal flagged YcgC. Such an observation should be described explicitly in the text to avoid potential confusion, especially if the K52R and K62R mutants don't migrate as fast. This type of shift by 1 or 2 Ac groups is quite large.*

We thank the reviewers for their insightful suggestions. The major concern was the observation of the 5-kDa downshift of RutR after the treatment of either YcgC or CobB, as it raised the possibility that YcgC might act as a protease that cleaved off the acetylated lysines of RutR. To explicitly address this concern, we have performed a series of assays. First, we recovered the downshifted RutR after YcgC treatment (e.g., Figure 2) and performed N-terminal Edman sequencing. We identified two versions of cleaved RutR: the major species involves deletion of 14-aa (N-MTQGAVKTTGKRSR), and the minor species involves deletion of 11 aa (N-MTQGAVKTTGK) from the N-terminus of RutR. Because the identified acetylated K52 and K62 are quite distant from the N-terminus, this N-terminal cleavage cannot account for YcgC’s KDAC activity, although it still did not rule out YcgC’s protease activity. Second, we sequenced the downshifted RutR after CobB treatment and found the same cleaved RutR missing the 14-aa at the N-terminus. Since CobB is a NAD^+^-dependent Class III sirtuin without any known protease activity, these results suggest that the RutR cleavage is intrinsic to RutR autoproteolysis. Third, to test whether RutR carries any protease activity, intact RutR proteins were heat-denatured and then incubated with either YcgC or CobB under the same deacetylation conditions. It became clear that denatured RutR could no longer downshift, while YcgC and CobB could still significantly deacetylate denatured RutR as compared with the untreated RutR. Finally, we examined the role of specific lysines in RutR’s proteolytic activity. As suggested by the reviewers, we created K52R, K62R and K52/62R mutants in RutR and tested them together with all the K-to-Q mutants in YcgC deacetylation reactions. On the basis of the ratios of cleaved/intact RutR in each reaction, we observed that when K62 was deacetylated in either K52Q or K52R, the proteolytic activity of these mutated RutR was almost as strong as WT RutR. In contrast, K62Q, K62R, and the two double mutants showed significantly lower proteolytic activities. Taken together, the above results confirmed YcgC’s KDAC activity and revealed that the autoproteolytic activity of RutR is dependent on deacetylation of K52/62 with K62 playing a major role. Because these new results are both important and interesting, we added a new Figure 3 and supplemental figures (Figure 2—figure supplement 4, Figure 3—figure supplement 2) to fully describe our new observations.

*3) Another possibility is to mutate the conserved Ser residues in YcgC and show that the mutant loses deacetylase activity. There are only five conserved Ser residues and mutating them would not be a huge effort.*

We thank the reviewers for this insightful suggestion. As the reviewers expected, once these five conserved Ser residues were mutated in YcgC, the S8/10/73/77/200A mutant could no longer deacetylate RutR (see new Figure 3). To further pinpoint the essential Ser for the KDAC activity, we incubated WT YcgC with hydrolase inhibitor AEBSF and the following MS/MS analysis revealed that Ser200 was the only residue covalently labeled with AEBSF (See new Figure 3). This result was further validated by the observation that S200A mutation completely abolished YcgC’s KDAC activity (Left panel; new Figure 3). We have now added these new results to the main text and to the new Figure 3.

*4) For Figure 1, the experimental details provided in the main text and response letter are still not clear. Did the authors run "Coo. Stain" on the samples and then use the same sample for the deacelylation reaction or did they run "Coo. Stain" and deacelylation with equal amounts of the two aliquots? If the latter is the case, the samples after the deacelylation reaction should be subsequently stain by a Western-compatible dye followed by anti-acetyllysine antibody.*

In the previous figure, one aliquot was Coomassie stained as loading control to show that equal amounts of RutR were loaded to each lane. Another aliquot was deacetylated and then WB by anti-acetyllysine antibody. To further clarify this, after the deacetylation reaction, RutR was divided into two aliquots of equal amounts: one aliquot was then subjected for Coomassie stain, and the other was western blotted by anti-acetyllysine antibody. We have also clarified this point in the manuscript accordingly (subsection “Screen new KDAC using the *E. coli* proteome microarray”, second paragraph).